

# Genesis
## Project Genesis a Platform for Automating IT Processes

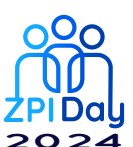

**Autors**: Dawid Rymarczyk · Jakub Wiraszka · Jakub Andrzejewski · Marek Kędzia

**Supervisor:** Natalia Piórkowska

### Abstract

The Genesis Project was developed as part of a Team Project(aka. ZPI) to simplify and automate the creation of IT environments for new projects. The goal of the initiative is to reduce the time required to transition from an idea to the development phase by eliminating repetitive tasks. The project is based on integration with external APIs and utilizes Terraform to automatically generate infrastructure code.

Key functionalities include cloud environment creation, integration with communication tools, and project management platforms. The result is a reduction in manual errors and a significant acceleration of project creation processes, which is particularly beneficial for small businesses and IT enthusiasts.

## 1   INTRODUCTION

The Genesis project was created in response to the challenges associated with time-consuming and repetitive processes involved in building IT infrastructure. The traditional approach to starting new projects, even those with basic complexity, requires numerous manual steps and configurations, significantly prolonging the time needed to transition from concept to implementation. This issue particularly affects small companies and IT teams that aim to launch projects quickly but are constrained by limited resources.

The goal of the Genesis project was to develop a tool that automates the process of creating a working environment by integrating with external APIs and using Terraform [3] to generate infrastructure code [4]. From a business perspective, the project aimed to provide users with a solution that shortens the time required to launch new projects and reduces the risk of manual errors, ultimately leading to faster time-to-market and resource optimization.

The objectives set for the team included integrating with key tools (Discord, ClickUp, GitHub, AWS [6]), building a flexible web application, and ensuring seamless creation of cloud infrastructure. The expected technical benefits included automated, rapid deployments and improved team efficiency by eliminating manual, repetitive tasks.

## 2   RELATED WORK

The project aimed to utilize modern technologies such as React with TypeScript on the frontend and NodeJS on the backend, hosted in an AWS environment. Key objectives included integration with external APIs, such as Discord, ClickUp, and GitHub, enabling full automation of processes—from creating communication environments to managing projects.

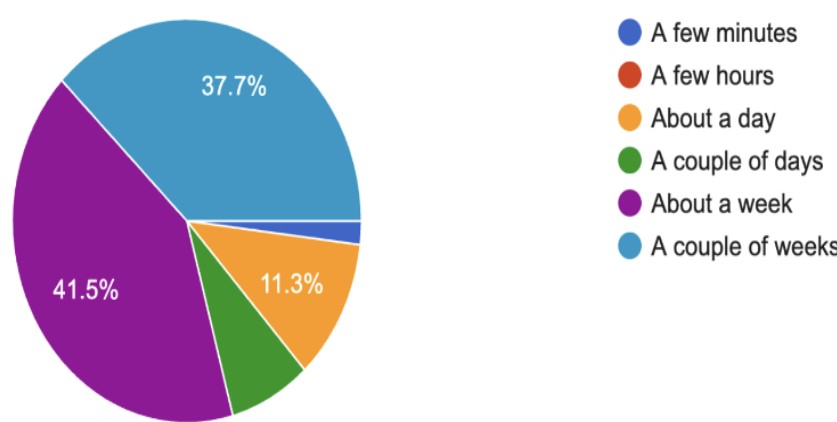

Figure 1: Total time spent on implementing project environment - survey

During the project preparation phase, we conducted a survey asking freelancers and small companies how much time they typically spend creating a working environment for a new project. The results were surprising. Participants indicated that this process takes from one week to even several weeks to fully prepare a working environment for a new project.

Before starting the project, we analyzed existing solutions and technologies for automating IT infrastructure creation. However, we could not find similar applications. Regarding project infrastructure, tools like Ansible [1] and Puppet [5] are available on the market to support deployment configuration automation. Still, they require advanced technical knowledge and are not fully integrated with a no-code approach. Genesis stands out by using Terraform to automate infrastructure building while offering an intuitive interface for users with basic technical skills.

| Aspect | Ansible | Puppet | Genesis |
|---|---|---|---|
| **Technology Base** | Built using Python, uses YAML | Built using Ruby, uses Puppet DSL | Built using TypeScript, uses HCL |
| **Communication** | Relies on SSH | Follows a client-server model using HTTPS communication | Communicates via REST APIs |
| **Agent** | Agentless | Requires an agent | Agent helps with infrastructure management |
| **Ease of Use** | Limited Windows support, moderate complexity | Complex syntax and steep learning curve | Easy to use, minimal technical knowledge required |
| **Error Potential** | Prone to errors due to manual configurations and playbook syntax issues | High potential for errors from complex scripts and agent dependency | Error-free due to predefined templates |

Table 1: Comparison of Ansible, Puppet, and Genesis technologies.

Before creating Genesis, we identified a niche in the market that existing solutions were failing to address. While tools like Ansible and Puppet are powerful, when it comes to automatization, they come with significant challenges such as steep learning curves, complex setups, and limited usability for less technical teams.

Genesis was designed to fill this gap by offering a modern, easy-to-use solution tailored for teams seeking simplicity. Built using widely adopted technologies like TypeScript and HCL, Genesis provides seamless communication via REST APIs and an intelligent agent that simplifies infrastructure creation. Genesis has user-friendly design that minimizes the need for extensive technical expertise, making

it accessible and efficient for a broader range of users. By targeting this underserved niche, Genesis delivers a unique value proposition in the infrastructure creation and automatization landscape.

The project faced constraints of time (limited to one academic semester), a small team of four members, potential API compatibility issues, the need to learn new technologies, and costs associated with AWS services.

## 3   RESULTS

### 3.1   Implemented Functionalities

The Terraform code generator enables the automatic creation of advanced cloud infrastructure in AWS, supporting services such as ECS, ELB, ECR, VPC, AWS Cognito, CloudFront, and S3. This generator reduces manual errors and saves time. The project facilitates repository creation, automatic addition of Terraform configuration files, and CI/CD pipeline setup. Users can manage projects, create tasks, assign members, and monitor progress through integration with ClickUp. The tool also supports the creation of Discord servers with predefined channels for team communication.

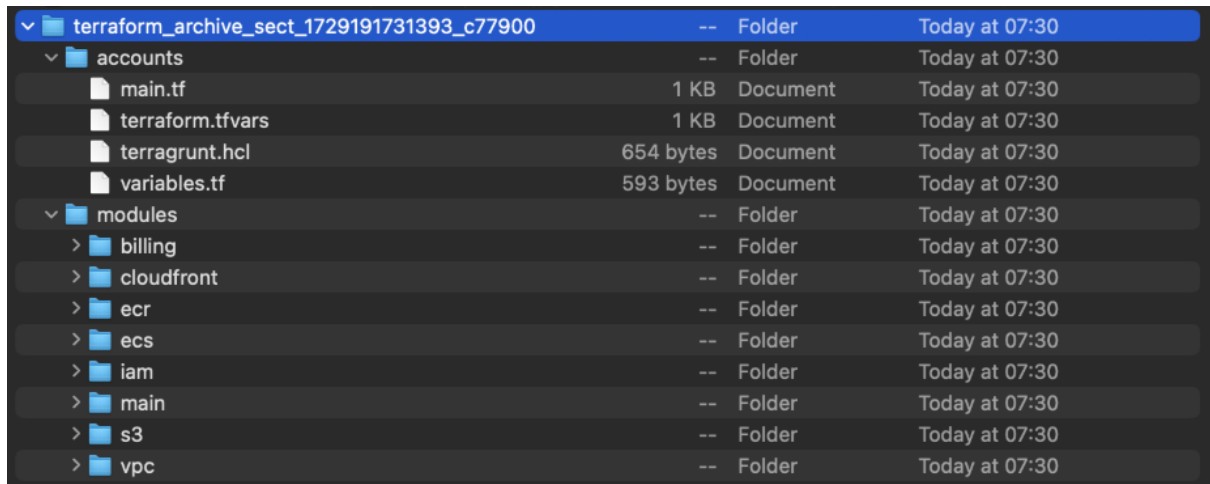

Figure 2: Ready-to-use and auto-generated infrastructure

Additionally, the project integrates with AWS Lambda, allowing the frontend to retrieve real-time updates on configured resources. AWS DynamoDB [2] is used as the primary database, and its non-relational nature and scalability enable fast operations and real-time data storage.

### 3.2   Business and Technical Objectives

**Business Benefits:** The implemented solutions have increased operational efficiency by reducing the time required for key processes, which will enhance the company's overall performance. The new functionalities enable better alignment of services with customer needs, allowing even users with limited technical expertise to leverage the application. Additionally, cost optimization was achieved through the automation of processes in line with best practices, contributing to more efficient resource management.

**Technical Aspects:** The use of modern technological solutions ensured the flexibility and generic nature of the system. The application was developed following best programming practices, with a strong emphasis on the DRY principle. The application is highly generic and presents scenarios where the frontend is controlled by the backend. This means that adding a new resource to the application would require changes only on the backend, to be more precise, creating SDKs to facilitate communication with external API services.

Data protection mechanisms were enhanced through the use of cloud services, contributing to compliance with current security standards. Optimized integration with existing systems enabled seamless data flow between various applications. The project's infrastructure was designed to be scalable to meet the growing demands of the company. The application is highly scalable, relying on cloud-based services that allow for automatic load adjustment in case of increased user traffic, ensuring consistently high-quality user experience.

## 3.3 Data and Success Metrics of the Project

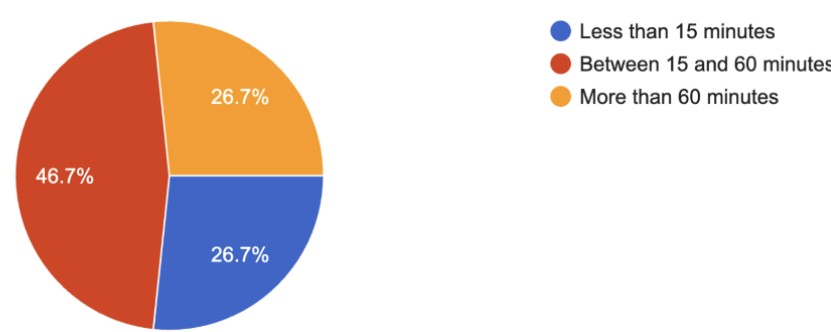

Figure 3: Outcomes after using Genesis Application

Operational efficiency has been significantly enhanced through a notable reduction in process completion times. The automation of key activities has accelerated project execution.

We conducted a survey among our beta testers to determine the average time they spent creating project's IT environment. The results were impressive. Most testers reported spending less than one hour to fully configure their environments.

This is a remarkable achievement, considering that without the Genesis application, setting up an IT environment typically required anywhere from a week to several weeks. This drastic improvement highlights the transformative potential of our tool in streamlining and optimizing IT workflows.

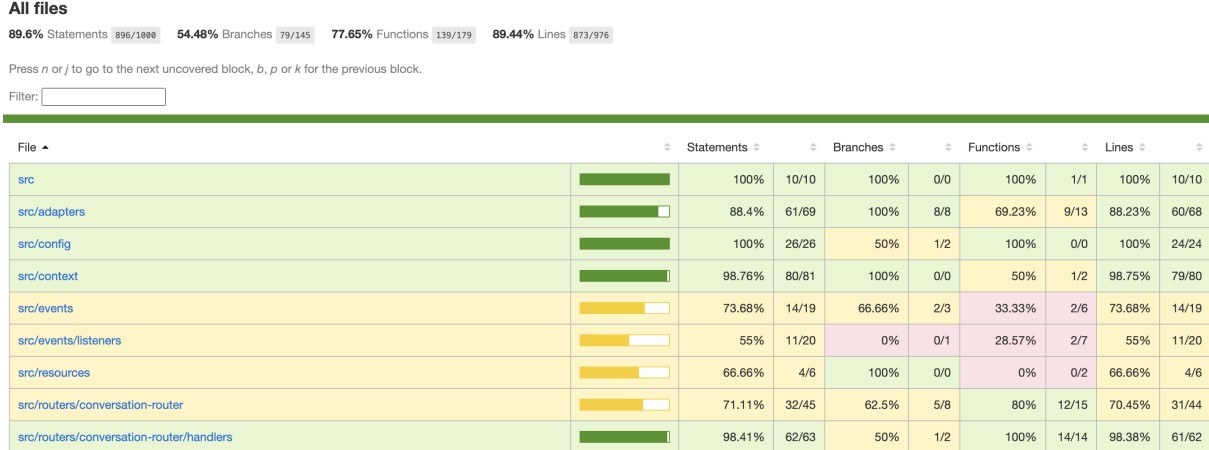

Figure 4: Code coverage

The system successfully passed all planned tests including unit tests, integration tests and comprehensive end-to-end (E2E) tests, ensuring the application's stability. This achievement reflects the high quality of the software. By automating processes, the risk of manual errors has been significantly reduced, leading to more stable and reliable deployments. These improvements underline the system's robustness and its potential to deliver consistent, almost error free performance.

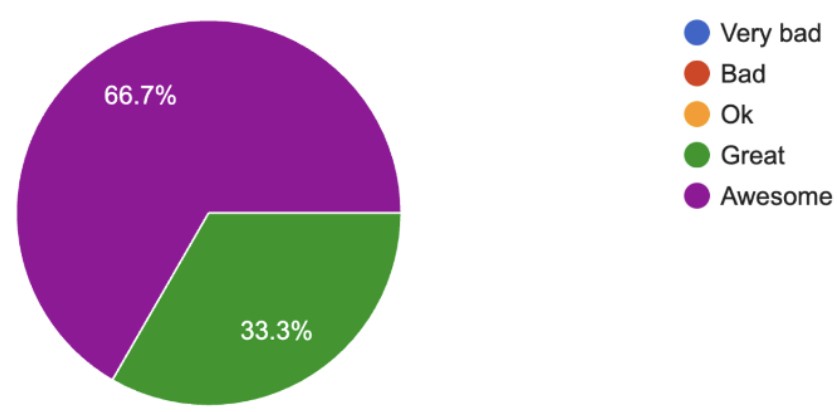

Figure 5: Genesis Feedback

So far, the feedback from beta testers has been overwhelmingly positive, which indicates that the system is effectively meeting user needs and expectations. From a business perspective, this suggests a strong potential for market adoption and highlights the system's capability to deliver tangible value to organizations by streamlining processes and enhancing efficiency.

## 3.4 Streamlined Deployment and User-Centric Benefits

The deployment of the Genesis project leverages a CI/CD process to automate and streamline the team's workflow. The system operates with two main branches: `develop`, managing the development environment, and `main`, representing the production environment. Automatic deployments are triggered when a pull request (PR) is merged into the respective branch, enabling a seamless transition of changes to the production environment hosted on AWS.

To manage project environments, Genesis uses AWS Elastic Beanstalk, ensuring an efficient and scalable hosting platform that simplifies configuration and application monitoring.

The benefits of the Genesis project encompass a wide range of conveniences. For technical users, these include primarily the acceleration of IT infrastructure creation, the ability to integrate with AWS cloud without requiring advanced technical knowledge, and the reduction of manual errors in environment configuration. Companies and startups can rely on simplified project management through integration with tools such as ClickUp and GitHub, translating into time and resource savings, especially for smaller businesses and freelancers. Developers gain the ability to focus on application development rather than infrastructure configuration, as well as the automatic creation of development and production environments. Finally, project teams benefit from easier communication and organization of work thanks to integration tools and intuitive project management within a single centralized system.

## 4   CONCLUSIONS

The Genesis project has delivered a transformative tool for automating IT project environment creation, significantly enhancing efficiency and accuracy. By integrating external services such as GitHub, Discord, and ClickUp and utilizing Terraform for automated infrastructure generation in AWS, the tool drastically reduced the time required for environment setup. Based on user surveys and beta tests, the average configuration time dropped from several weeks to under one hour-a reduction of over 90%.

From a business perspective, Genesis simplifies and accelerates IT project launches, eliminating manual errors by automating complex processes. The infrastructure generated by Genesis is error-free, ensuring reliability and consistency.

A key technical achievement is the seamless integration of Terraform code generation with robust API connections, enabling the creation of comprehensive IT infrastructures in minutes. This flexibility, combined with its ability to support scalable and secure deployments, positions Genesis as a rare solution tailored for smaller entities that often lack advanced IT capabilities.

In summary, the Genesis project successfully addressed its core objectives by demonstrating significant time savings, enhanced process reliability, and operational efficiency.

## 5   FUTURE DEVELOPMENT DIRECTIONS

In the project, we have identified two key areas for development: the implementation of a message queue (e.g., Amazon SQS, RabbitMQ or Kafka) and the application of machine learning (ML) based on the collected data.

Currently, communication between the backend and frontend is handled via REST API, which means synchronous, blocking data exchange. Implementing a message queue would enable a transition to asynchronous communication, allowing for non-blocking processing of large volumes of data, thereby increasing the scalability and efficiency of the system.

In the context of machine learning, we see significant potential in leveraging the collected data to create predictive models and perform analyses. Through the implementation of ML algorithms, the system could analyze user behavior patterns, predict resource infrastructure needs, and optimize automatic configurations. ML integration could also support decision-making automation and suggest optimal infrastructure solutions to users based on historical data and current trends.

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
