# OpenReview forum: "Project Genesis a Platform for Automating IT Processes"
_pwr.edu.pl/Wrocław_University_of_Science_and_Technology/2024/ZPI_Day — Wrocław University of Science and Technology 2024 ZPI Day Submission_

### Official Review · Reviewer_bb1m · 2024-12-06
**The project is interesting, but requires a lot of work to gain generality in the range of technologies used.**

**Confidence:** 5
**Significance Of Results:** 4
**Overall Quality:** 4

**Compliance With Template:**

5: Very High Quality – The article contains all the required sections, which are written in a very detailed, clear, and error-free manner. The structure is professional and meets expectations, and the content adheres to the highest substantive and formal standards.

**Description Of Results:**

4: High Quality – The results are described in detail and supported by usage examples or evaluations. The description is reliable but may lack full depth of analysis.

**Feedback On Consistency:**

Everything is coherent and logical, but the description would be better if it included an example clearly showing the subsequent steps of creating a work environment. It should also include the information collected before starting to work with this system.

**Potential For Development:**

The article indicate possibilities for further work and practical applications.

**Project Nature Evaluation:**

The project exhibits characteristics of an engineering work. It addresses important issues in ensuring the rapid development of an efficient working environment.

**Technical Language Precision:**

4: High Quality – The language is appropriate for a technical report. Terminology is used correctly, and statements are precise, with only minor shortcomings that do not affect the overall clarity.

---

### Official Review · Reviewer_HfXs · 2024-12-08
**IT project prepared correctly.**

**Confidence:** 5
**Significance Of Results:** 5
**Overall Quality:** 5

**Compliance With Template:**

5: Very High Quality – The article contains all the required sections, which are written in a very detailed, clear, and error-free manner. The structure is professional and meets expectations, and the content adheres to the highest substantive and formal standards.

**Description Of Results:**

5: Very High Quality – The results are described in detail, clearly and comprehensively, supported by thorough evaluation, analysis, and convincing usage examples. The description meets the highest substantive standards.

**Feedback On Consistency:**

The analysis of the problem, presentation of results and conclusions are coherent and logical

**Potential For Development:**

The article indicates possibilities for further work or practical application of its results.

**Project Nature Evaluation:**

Both the level of usability, the applied technical methods and technological solutions have the characteristics of engineering work.

**Technical Language Precision:**

5: Very High Quality – The language is entirely appropriate for a technical report. All terms are used correctly and precisely, and the style is professional, clear, and coherent, without any errors or ambiguities.

---

### Official Review · Reviewer_5VpY · 2024-12-08
**The project is interesting and useful, but the report includes some claims which seem to be not backed up by the presented data.**

**Confidence:** 4
**Significance Of Results:** 3
**Overall Quality:** 3

**Compliance With Template:**

4: High Quality – The article contains all the required sections, which are well-written and substantively correct, although minor errors or shortcomings may be present. The overall structure is clear and coherent.

**Description Of Results:**

3: Average Quality – The results are described with moderate detail. Some examples or evaluation elements are present but insufficiently developed or incomplete.

**Feedback On Consistency:**

Problem analysis, presentation of results, and conclusions are consistent and logical. Addiotional remarks apply:

>>> 1. Language problems: not detected

>>> 2. Presentation problems:
2.1 Fig 4: title is too general
2.2 Fig 5: title is too general
2.3 The diagram of the proposed system is not presented

>>> 3. Other problems:
3.1 Autors write: "The implemented solutions have increased operational efficiency by reducing the time required for key processes, which will enhance the company’s overall performance."
This claim is not backed up by the presented data.

3.2 Authors write: "In summary, the Genesis project successfully addressed its core objectives by demonstrating significant time savings, enhanced process reliability, and operational efficiency."
This claim is not backed up by the presented data. How process reliability and operational efficiency were defined, measured and compared?

3.3 Authors write: "The new functionalities enable better alignment of services with customer needs, allowing even users with limited technical expertise to leverage the application. Additionally, cost optimization was achieved through the automation of processes in line with best practices, contributing to more efficient resource management."
This claim is not backed up by the presented data. How costs and "customer needs" were defined, measured and compared?

===EOT===

**Potential For Development:**

Yes.

**Project Nature Evaluation:**

Yes, the project exhibits characteristics of an engineering work, with high level of utility, application of technical methods, and technological solutions.

**Technical Language Precision:**

4: High Quality – The language is appropriate for a technical report. Terminology is used correctly, and statements are precise, with only minor shortcomings that do not affect the overall clarity.

---

### Decision · Program_Chairs · 2024-12-10

Accept (Poster)